- Reconstructing Sea Level Variability at the Ieodo Ocean Research
- Station (1993–2023) Using Artificial Intelligence, Machine Learning,
- and Reanalysis Integration
- MyeongHee Han and Hak Soo Lim\*
- Marine Natural Disaster Research Department, Korea Institute of Ocean Science and Technology, Busan,
- Republic of Korea
- \*Corresponding author: Hak Soo Lim (hslim@kiost.ac.kr)

### Abstract

This study presents a comprehensive approach for reconstructing a high-quality, continuous 14 monthly sea level time series at the Ieodo Ocean Research Station (IORS) from 1993 to 2023 15 using advanced artificial intelligence (AI) and machine learning (ML) models. After applying quality control to the in-situ KIOST data, including inverse barometric effect correction, 3σ 16 17 filtering, and a 75% data coverage threshold, we validated trends using nearby PSMSL tide 18 gauges and four ocean reanalysis datasets (CMEMS, GLORYS, ORAS5, HYCOM). The trend 19 analysis showed a higher rate of sea level rise from in-situ data (4.94 mm/yr, Oct 2003-Dec 20 2023) compared to satellite and model-based estimates (e.g., CMEMS: 3.53 mm/yr, Jan 1993-21 Dec 2023), suggesting localized sea level rise in the East China Sea. Initial gap-filling used 22 statistical models such as harmonic regression and regression-based climatology. A blended 23 approach combining climatology and trend components achieved the best accuracy (RMSE 24 ~0.056 m,  $R^2 = 0.688$ ). We then implemented various AI/ML models through an Iterative Imputer framework. Ensemble models (e.g., XGBoost) performed perfectly after 2003 but did 25 26 not generalize well before 2004. Deep learning models like LSTM and GRU effectively captured seasonal and nonlinear patterns post-2003, with LSTM achieving RMSE = 0.023 m 27 28 and  $R^2 = 0.95$ . Time series models Prophet and SARIMA-SIN successfully reconstructed the 29 full time series, with SARIMA-SIN estimating the highest trend (5.61 mm/yr). Multiple linear 30 regression using reanalysis data served as a baseline, but AI/ML models outperformed it in both 31 accuracy and generalization. This study provides a reproducible, interpretable, and physically 32 consistent framework for reconstructing sea level variability in semi-enclosed coastal seas.

33

Keywords: Sea Level Variability, Ieodo Ocean Research Station, AI-based Imputation,

Timeseries Reconstruction, East China Sea

## 36 1. Introduction

| 37 | Long-term sea level observations are critical for monitoring regional oceanographic and           |
|----|---------------------------------------------------------------------------------------------------|
| 38 | climatic changes, particularly in coastal and marginal seas where variability is amplified by     |
| 39 | topographic and atmospheric forcing (Hamlington et al., 2020; Cazenave and Moreira, 2022).        |
| 40 | The Ieodo Ocean Research Station, located at the intersection of the East China Sea (ECS) and     |
| 41 | the Southern East Sea (Sea of Japan), plays a pivotal role in observing regional sea level        |
| 42 | variability and marine environmental conditions (Han, 2020; Byun et al., 2021). As a              |
| 43 | strategically important in-situ platform, its various interval sea level records provide valuable |
| 44 | information for understanding the dynamics of the Kuroshio Current system, East Asian             |
| 45 | monsoon variability, and climate change-driven sea level rise (Ha et al., 2019; Xu et al., 2015;  |
| 46 | Chang and Oey, 2011).                                                                             |
|    |                                                                                                   |

However, long-term in-situ observations are often interrupted by equipment failure, maintenance issues, or extreme weather conditions such as typhoons (Adebisi et al., 2021). These disruptions lead to temporal gaps in the observational records, which hinder the detection of trends, reconstruction of seasonal cycles, and validation of satellite and model-based products (Beguería et al., 2019). In regions like the ECS, where strong seasonal and interannual signals are present, accurate and realistic imputation of missing data is essential for scientific and operational applications (Lin et al., 2020; Han, 2020).

| 54 | Various methods have been developed to address missing data in oceanographic and climate          |
|----|---------------------------------------------------------------------------------------------------|
| 55 | time series (Kolukula and Murty, 2025; Lee et al., 2022). Traditional approaches include linear   |
| 56 | interpolation, monthly climatological averages, and harmonic regression models (Schlegel et       |
| 57 | al., 2019; Risien et al., 2022; Okkaoğlu et al., 2020; Arguez and Applequist, 2013). More         |
| 58 | recently, advanced statistical and machine learning techniques have been proposed for gap-        |
| 59 | filling, including Gaussian process regression, Kalman filtering, and neural networks (Vance      |
| 60 | et al., 2022; Wenzel and Schröter, 2010; Wang, 2023). While these methods offer improved          |
| 61 | flexibility and accuracy, they often require dense observations or training data, which may not   |
| 62 | be feasible in long-term sparse records (Lee et al., 2022; Sarafanov et al., 2022; Park et al.,   |
| 63 | 2023). Interpretable and statistically robust methods remain essential for operational and        |
| 64 | historical datasets such as IORS (Han and Lim, 2024; Han, 2020).                                  |
| 65 | This study focuses on imputing, filling, and predicting gaps in the monthly sea level data at the |
| 66 | IORS over 1993-2023. We evaluate and compare three regression-based approaches: (1)               |
| 67 | harmonic regression with annual and semiannual cycles plus a linear trend (Okkaoğlu et al.,       |
| 68 | 2020), (2) a regression-based monthly climatology model with a trend using calendar year          |
| 69 | month dummy variables (Hyndman and Athanasopoulos, 2018), and (3) a pure climatology-             |
| 70 | plus-trend approach based on aggregated monthly means and linear fitting (Brunetti et al.,        |
| 71 | 2014). Furthermore, we propose a realistic blending method that optimally combines harmonic       |

and climatological components to minimize reconstruction errors.

| 73 | Also, we implemented an ensemble of statistical and machine learning (ML) models to                  |
|----|------------------------------------------------------------------------------------------------------|
| 74 | reconstruct missing monthly values at the IORS (Tong and Li, 2025; Bochow et al., 2025).             |
| 75 | These models were selected to span a broad range of algorithmic families, including ensemble         |
| 76 | tree-based methods (e.g., XGBoost, Random Forest, LightGBM, AdaBoost) (Niazkar et al.,               |
| 77 | 2024; Wu et al., 2024; Gan et al., 2021; Xiao et al., 2019), regularized linear models (Lasso,       |
| 78 | Ridge) (Pan et al., 2025), proximity-based models (K-Nearest Neighbors) (Latif et al., 2024),        |
| 79 | and neural networks (LSTM, GRU) (Sun et al., 2020; Tumse and Alcansoy, 2025). Each model             |
| 80 | was trained to impute missing values using only observed sea level data from the original            |
| 81 | dataset, applying the IterativeImputer framework with consistent hyperparameters for                 |
| 82 | comparability (Ramirez et al., 2023). Complementing these were two timeseries specific               |
| 83 | models: Facebook's Prophet (Elneel et al., 2024), which decomposes time series into seasonal         |
| 84 | and trend components, and SARIMA (Sun et al., 2020), which captures both non-seasonal and            |
| 85 | seasonal autocorrelation structures. All models were evaluated using root mean square error          |
| 86 | (RMSE) and coefficient of determination (R <sup>2</sup> ), calculated against observed (non-missing) |
| 87 | values (Boursalie et al., 2022; Siddig et al., 2021).                                                |
|    |                                                                                                      |

To contextualize the reconstructed series, we further compared the IORS observations with 89 monthly sea level data from four global ocean reanalysis products—Copernicus Marine

- Environment Monitoring Service (CMEMS), Global Ocean Physics Reanalysis System
- (GLORYS), Ocean Reanalysis System 5 (ORAS5), and Hybrid Coordinate Ocean Model
- (HYCOM)—over 1993–2023 (Han et al., 2024; Long et al., 2023; Jin et al., 2023b; Cummings
- and Smedstad, 2014). Linear trends were estimated and visualized to assess the consistency
- and fidelity of in-situ observations relative to reanalysis-based records.
- Finally, we extended the imputation framework to a multivariate setting by incorporating these
- reanalysis datasets as auxiliary predictors, enabling more robust and physically informed gap-
- filling. By leveraging both statistical and AI/ML-based techniques, this study provides a
- transparent, reproducible framework for reconstructing realistic monthly sea level time series
- in data-sparse coastal environments.