# Peer review of "Reconstructing Sea Level Variability at the Ieodo Ocean Research"

_Earth System Science Data, 2025_

## Author Comment (AC1)

The authors made an ambitious effort to reconstruct sea level variability from Ieodo Ocean Research Station using various models, AI/ML tools, and observations. I can appreciate the authors' diligent and thoughtful analysis. A few suggestions that I hope to make this manuscript better.

- We sincerely appreciate your careful and constructive comments, which have greatly helped us improve the clarity and quality of the manuscript. I have addressed all of your questions and suggestions based on the revised manuscript. Unfortunately, I am unable to upload the revised manuscript directly here, but I would be happy to provide further clarification or additional information if needed. Please do not hesitate to let me know if you have any additional questions or comments.

First, throughout the manuscript, the KIOST in-situ data, KIOST sea level time series, IORS data, KIOST tide gauge, and IORS observation were used interchangeably. Do authors mean the same data set at IORS station maintained by the KIOST? Are there any other data sets from the KIOST? Please use the same acronym consistently to represent the data.

- Thank you for pointing this out. We confirm that all these terms refer to the same insitu sea level dataset recorded at the leodo Ocean Research Station (IORS), maintained by the Korea Institute of Ocean Science and Technology (KIOST). In the revised manuscript, we now consistently use "IOSR" to refer to this dataset throughout.

Second, the main parameter compared is the linear "trend" from various models and observations, but I don't see any regression plot on the manuscript to indicate how the trend is computed.

We appreciate this important suggestion. To clarify, linear trends were computed using ordinary least squares regression of monthly sea level anomalies against time (in months). To improve transparency, we have added a new figure (Figure 8) showing both the monthly time series and their corresponding regression lines for IORS (with trends over 2003–2023 and 2004–2023), CMEMS, GLORYS, ORAS5 (all over 1993–2023), and HYCOM (over its valid range, 1994–2023). This revised figure and caption explain the specific time windows used for each regression.

Third, at the beginning of section 3, a significant upward trend of approximately 4.94 mm/yr from October 2003 to December 2023 and 5.43 mm/yr from January 2004 to December 2023 were reported. Is this from two different observation data sets? Why on the same location, only a few months of difference over a long 20 year period will have a different trend?

- Both trends were computed from the same IORS dataset. The 4.94 mm/yr trend

corresponds to October 2003 to December 2023, beginning with the first available observation, while the 5.43 mm/yr trend covers January 2004 to December 2023, aligning with full calendar years. The difference (~0.5 mm/yr) reflects the sensitivity of linear regression to initial values, especially relatively high sea levels in October and November 2023, which exert downward leverage on the trend estimate. As a result, the earlier start yields a slightly lower (less steep) slope. This explanation has been incorporated into the manuscript, and both regression lines are shown in Figure 8 to illustrate the impact of the starting point. To ensure consistent comparison across datasets, Table 5 has been revised to report trends over the common period January 2004 to December 2023 for all satellite and reanalysis products (CMEMS, GLORYS, ORAS5, HYCOM), while also retaining the full-period trends (1993–2023 or 1994–2023) for completeness.

Fourth, there are a few important figures that I really appreciate, i.e. Figure 5, 7 & 9. But too many lines with various legends get on top of each other. I cannot tell which one is which. For example, Figure 9 has 15 lines on top of each other. I suggest making the figures and legends larger and maybe make separated plots.

Thank you for this helpful suggestion. We revised the figures as follows:

- Figure 5: We introduced vertical offsets and reordered the legend by distance to the IORS.
- Figure 7: We adopted a similar offset style and added trend values to the legend.
- Figure 9: We revised the figure by applying vertical offsets to the reconstructed sea level time series from each model to reduce visual overlap. Models are labeled with both offset and linear trend values in the legend

These changes significantly improve figure readability, and we appreciate your guidance.

Other comments:

Line 16: What is KIOST? Acronym first appearance

- We clarified the first mention of KIOST as "Korea Institute of Ocean Science and Technology (KIOST)" in Line 112 and removed the redundant mention in Line 16.

Line 17: What is PSMSL? Never explained.

- PSMSL is now defined as "Permanent Service for Mean Sea Level (PSMSL)" in Lines

**17–18.**

Line 21-22 "Initial gap-filling used statistical models such as harmonic regression and regression-based climatology".  $\rightarrow$  What are you trying to say?

- We have rewritten the sentence as "Initial gap-filling of the in-situ IORS sea level data was conducted using statistical models, including harmonic regression and regression-based climatology." in Lines 22-23.

Line 21-24: Unclear. Please rewrite.

We have rewritten the sentence as "Initial gap-filling of the in-situ IORS sea level data was conducted using statistical models, including harmonic regression and regression-based climatology. A blended approach, integrating climatological cycles with a linear trend, yielded the highest accuracy when validated against observed (non-missing) data (RMSE ≈ 0.056 m; R2 = 0.688)." in Lines 22-25.

Line 24-26 "Ensemble models (e.g., XGBoost) performed perfectly after 2003 but did not generalize well before 2004."  $\rightarrow$  This statement seems contradictory. Do you mean after 2003 is good, before 2004 is bad? How about 2003? Do you know why after 2003 is better? Can you speculate a reason? You don't have observation prior the 2003, how do you know prior 2004 is bad?

- We have rewritten the sentence as "We then implemented various AI/ML models through an Iterative Imputer framework. Ensemble models (e.g., XGBoost) accurately reproduced IORS observations after October 2003 but generated unrealistic values before this period, likely due to overfitting and limited ability to extrapolate beyond the training data range." in Lines 25-29.

Line 76 XGBooster? Acronym first appearance.

- We have corrected to "Extreme Gradient Boosting (XGBoost)." in Line 80.

Line 81 IterativeImputer  $\rightarrow$  Iterative Imputer

- We have corrected to "Iterative Imputer" in Line 85 and the main text; we retain IterativeImputer as the code reference in the Methods section. Line 101 Figure 1 is not mentioned in the text.

- We now cite Figure 1 in the main text in Line 45.

Line 101 Figure 1, What are KAN, LUS, SEO, FUR, NAK, NAS, NAH on the figure? I know they are on table 2, but ..

- The figure caption now includes "KAN, LUS, SEO, FUK, NAK, SIM, NAS, and NAH represent KANMEN, LUSI, SEOGWIPO, FUKUE, NAKANO, SIMA, NASE III, and NAHA" in the caption of Figure 1 in Line 105.

Line 113-114 Can you give a statistics/ percentage of how many data points were missing and data gaps?

We have added "Of the 372 expected monthly records from 1993 to 2023, 189 entries (50.8%) were missing. From the start of continuous observations in October 2003 to December 2023, 60 out of 243 months (24.7%) contained missing data." In Lines 121-123.

**Line 130: What is Epa, 2000?**

- We have corrected to "(U.S. Environmental Protection Agency, 2000)" in Lines 139-140.

Lien 137: Figure 2: What Good Quality Flag has data only @  $\sim$ 2004 - 2008? Why do the "quality flags" have variations? What do you mean 'flag'?

- We have revised the caption to explain that gray dots represent 10-minute data flagged as "good quality." Their prominence around 2004–2008 reflects changes in instrumentation and overplotting in Lines 147-157.

Line 163: Please explain the variables in Eqn (2).

- We have explained the variables in Lines 181-186.

Line 186: Please explain the variables in Eqn (4). What is Alfa\_o Alfa\_a?

- We have explained the variables including  $\alpha_0$  and  $\alpha_1$  in Lines 209-211.

Line 349 IterativeImputer → Iterative Imputer

- We have corrected to Iterative Imputer in Line 373.

Line 370-372 "a significant upward trend of approximately 4.94 mm/yr from October 2003 to December 2023 and 5.43 mm/yr from January 2004 to December 2023."  $\rightarrow$  Can you explain how these two numbers of the upward trend were obtained. Did you make a regression fit? Can you show them in a figure? Why is it only a few months different (Oct 2003 vs Jan 2004) over a long period of 20 years that the upward trend has so much difference? If you want to compare the upward trend of IORS and other models, why don't you use the same period of time?

The trends were derived using ordinary least squares linear regression and are illustrated in Figure 8. Relatively high sea level values in October and November 2003 exert downward leverage on the regression line, resulting in a slightly lower trend estimate when this period is included (Lines 395-399). To ensure consistency across datasets, all trend comparisons in Table 5 use the period January 2004 to December 2023. Table 5 has also been added to explicitly demonstrate the influence of the initial months on the calculated trend values.

Line 375 Table 2, caption. "five PSML tide gauge stations " $\rightarrow$  There are seven stations listed on Table 2.

- We have corrected Line 403 in Table 2 caption to accurately indicate seven PSMSL tide gauge stations are included.

Line 475 Figure 7: This is a great figure which shows some AI/ML models can reproduce the IORS observations. Unfortunately, there are too many lines, I can not tell which one is which. Can you either reduce the number of lines or make separated plots?

- Figure 7 was redrawn using separated offset plots, improving clarity while preserving the information in Line 508.

Line 669- "Among AI/ML models, ensemble learners (e.g., 670 XGBoost, RandomForest) achieved perfect reconstruction metrics after September 2003 but failed to predict values in earlier periods".  $\rightarrow$  You don't have observation data prior to 2003, how do you know "failed to predict values in earlier periods"?

- We acknowledge that no observational data exist before October 2003. Our comment about ensemble models "failing to predict" earlier values refers to their unrealistic and flat outputs in this period, as seen in Figure 7. This was not a validation failure but a

lack of generalization in the absence of physical context. When supplemented with satellite/reanalysis predictors (Figure 9), these models performed well across the entire period (1993–2023).